# Improving the Accuracy of Predictive Models for Outcomes of Antidepressants by Using an Ontological Adjustment Approach

Hua Min [1,*] , Farrokh Alemi [1,2] , Christopher A. Hane [2] and Vijay S. Nori [2]

1 Department of Health Administration and Policy, George Mason University, Fairfax, VA 22030, USA; falemi@gmu.edu
2 OptumLabs, Eden Prairie, MN 55344, USA; christopher.hane@optum.com (C.A.H.); vijay.nori@optum.com (V.S.N.)
* Correspondence: hmin3@gmu.edu; Tel.: +1-703-993-5648

**Featured Application: Ontological hierarchies provide broader concepts that can be used to increase sample size for estimating the effect of treatment in rare conditions, thus improving the accuracy of the predictive models. Ontology adjustment can be used to prevent overfitting in big data analysis.**

**Abstract:** For patients with rare comorbidities, there are insufficient observations to accurately estimate the effectiveness of treatment. At the same time, all diagnosis, including rare diagnosis, are part of the International Classification of Disease (ICD). Grouping ICD into broader concepts (i.e., ontology adjustment) can not only increase accuracy of estimating antidepressant effectiveness for patients with rare conditions but also prevent overfitting in big data analysis. In this study, 3,678,082 depressed patients treated with antidepressants were obtained from OptumLabs® Data Warehouse (OLDW). For rare diagnoses, adjustments were made by using the likelihood ratio of the immediate broader concept in the ICD hierarchies. The accuracy of models in training (90%) and test (10%) sets was examined using the area under the receiver operating curves (AROC). The gap in training and test AROC shows how much random noise was modeled. If the gap is large, then the parameters of the model, including the reported effectiveness of the antidepressant for patients with rare conditions, are suspect. There was, on average, a 9.0% reduction in the AROC gap after using the ontological adjustment. Therefore, ontology adjustment can reduce model overfitting, leading to better parameter estimates from the training set.

**Keywords:** depression; antidepressant; predictive model; ontology; ontology adjustment; overfitting; rare conditions

## 1. Introduction

Depression is a common chronic mental illness that can affect thoughts, mood, and physical health. It is a complex disease, and the causes of it can range from genetics, other diseases, and medications to life events. Depression has been found to be associated with increased mortality risk in persons with chronic diseases including heart disease [1,2], end-stage renal disease [3], diabetes [4], and suicide in the depressed population [5]. Besides mortality risk, depression is also related to disability burden such as an increase in unemployment rate and a decrease of annual income [6] and family burdens [7], especially for pregnant women [8].

Antidepressants are used as first-line treatment for depression. However, current studies have shown that patients with depression do not have a satisfactory therapeutic outcome. It is important to design and implement outcome-based prescription for antidepressant studies. Large available administrative claims data are increasingly being used for healthcare utilization, pharmacovigilance, comparative effectiveness research, and other studies [9–11]. Our study examined the patterns of antidepressant use, and

other information readily available in claims data, to predict the effectiveness of antidepressants in different subgroups of patients. We created models to predict remission for the top 20 antidepressants including bupropion, citalopram, escitalopram, fluoxetine, and sertraline.

Overfitting is a common issue in model development, especially in big data analysis involving thousands of features [12,13]. Overfitting happens when model parameters predict random noise in the training data. The model looks accurate in the training dataset, but accuracy drops in test data, when the same noise is no longer present. The parameters of rare events are especially suspect. There are too few observations of these events to accurately set these parameters, leading to a deterioration of model performance in test datasets. If there is little difference in the accuracy of the model predictions in training and validation sets, then the model has not captured random noise, and therefore, its parameters for rare events can be more trustworthy.

Ontologies have been used in emerging data-driven science, including data mining and machine learning [14,15]. Ontologies can help to cluster large amounts of data into meaningful and manageable groupings. For example, Gene Ontology is widely used to cluster gene expression data obtained from microarray experiments [16–18]. Previous studies had also examined the use of International Classification of Disease (ICD) Ontology for more accurate billing [19–21]. However, few studies investigate the usage of ontology in model overfitting, especially in the big data analysis. Our study focused on filling in this gap. We investigated how to use ontological hierarchies to adjust the sample size of patients with various diagnoses, especially for rare diagnoses. In moving up the ontological hierarchy, one increases the sample size, as broader concepts have more observations. We also discovered that ontological adjustment can be used to overcome model overfitting in the prediction of outcomes for antidepressants using large available claims data.

## 2. Materials and Methods

### 2.1. Source of Data

This study utilized data from the OptumLabs® Data Warehouse (OLDW), which includes national deidentified claims data of more than 125 million privately insured individuals, for patients treated between 1 January 2001 and 31 December 2018 [22]. The dataset contains individuals of various ages (including Medicare Advantage beneficiaries) from all 50 states, with greatest representation in the Midwest and South US Census Regions [23].

### 2.2. Sample

Patients were included in the study if they met following criteria: (1) diagnosed with depression, (2) aged older than 13, (3) prescribed with at least one antidepressant, and (4) enrolled for at least 1 year prior to the start of the medication and 100 days after the start of the medication. In total, 3,678,082 patients were eligible for the study. The analysis was completed at the treatment episode level and not at the patient level, since each patient may try several antidepressants before finding success. There were 10,221,145 episodes of antidepressant use.

### 2.3. Measure of Antidepressant Effectiveness

Patient-reported remission was not available in the data and was estimated using a surrogate index composed of five variables: (1) the number of prior antidepressants, (2) duration of medication in the current episode, (3) whether the patient switched to another medication before 100 days, (4) whether the patient augmented their current antidepressant, and (5) whether the medication dose reached therapeutic levels [24]. Symptom remission was predicted from the patient's medical history (age, gender, diagnoses, medications, and procedures) one year prior to taking the first antidepressant. There was a total of 40,784 unique predictors.

*2.4. Methods of Ontological Adjustment*

Ontological adjustment (OA) was defined as using ontology hierarchies to adjust the sample size of patients with various diagnoses, especially for rare diagnoses. Our ontological adjustment method contained three steps: Step 1: set the threshold for rare conditions. Step 2: calculate the rank of occurrence. Step 3: apply the ontological adjustment.

Step 1: a diagnosis was considered rare, and subject to ontological adjustment, if it occurred in less than a threshold number of patients in the data. Thresholds of 1, 30, 100, or 300 were tested. The best threshold for the ontological adjustment was chosen if it resulted in the best balance between the sample size (i.e., number of episodes) and levels of climbing hierarchies (i.e., number of codes affected).

Step 2: when a diagnosis occurred multiple times for the same patient, it was treated as a different variable, using the rank of occurrence. For example, 1st, 2nd, 3rd, 4th, and more than 4 occurrences of diabetes (ICD code 250) occur at different rates in our population. Rank was used as a surrogate to reflect the severity of a disease or disorder. A patient who has been diagnosed with an ICD code more than 4 times (i.e., rank 4+) is more likely to have a more severe case than those who have been diagnosed only once (i.e., rank 1) in their medical history.

Step 3: the ontological adjustment was applied to those ranks which were less than the chosen threshold. Given a diagnosis code and rank, the study algorithm combined ranks first and, when all rank adjustments had been made, then adjusted the code itself by climbing its ontological hierarchy.

Figure 1 demonstrates examples of how our algorithm worked when the threshold for rare diagnosis was set to 100 cases. In the first example, the ICD code 25050 had 130, 140, 110, 110 episodes for rank 1, 2, 3, 4+, respectively. There was no ontological adjustment needed for those variables since all variables occurred for more than the threshold of 100 treatment episodes (see Figure 1a). For the ICD code 25030, rank 3 and rank 4+ were less than the threshold, so they were added together to meet the threshold (see Figure 1b). For the ICD code 25020, rank 2 and rank 3 were less than the threshold. Therefore, we added rank 2, 3, and 4+ together (40 + 10 + 110) (see Figure 1c). In the last example (Figure 1d), all ranks for the ICD code 25021 and 25022 were less than the threshold. In that case, we dropped the last digit of the ICD codes and moved to its parent ICD code 2502.

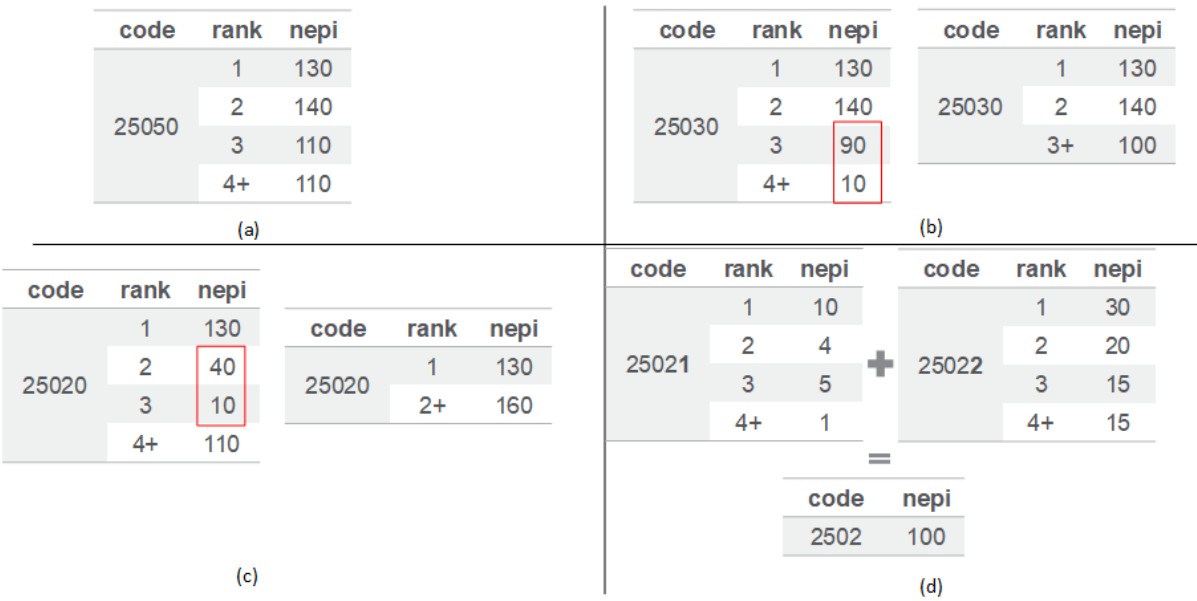

**Figure 1.** Examples showing rank and ontological adjustments. Observations less than 100 are considered rare. (**a**) no ontological adjustment for the ICD code 25050. (**b**) ontological adjustment for the ICD code 25030. (**c**) ontological adjustment for the ICD 25020 and (**d**) ontological adjustment for both ICD codes 25021 and 25022.

### 2.5. Measure of Accuracy of Estimated Treatment Effectiveness

For each of 20 most common antidepressants, the data were randomly partitioned into a training set (90%) and test set (10%). Naïve Bayes models were constructed to predict symptom remission after taking the antidepressant. Despite false assumption of independence of predictors in large sparse data such as the data in this study, naïve Bayes has been relatively accurate in predicting patient outcomes [25–28]. For each factor, including each diagnosis, a likelihood ratio for symptom remission was estimated from the training set. These likelihood ratios could be erroneous if estimated from few cases, in which case the overall accuracy of the Bayes model is affected. Accuracy of the Bayes models was measured through area under the receiver operating curve (AROC). The difference between AROC in the training and test dataset (called AROC gap) showed how much the model in the training set had captured random noise. If the gap was large, then model parameters (the likelihood ratios for symptom remission) were suspect because these estimates were obtained from a training set that had captured more noise.

## 3. Results

### 3.1. Threshold for Adjustments

Four different thresholds for rare diagnoses were tested for our dataset (i.e., 1, 30, 100, and 300). Figure 2 shows the percentage of ICD codes adjusted based on the different thresholds. For example, no codes needed to be adjusted for the threshold 1. However, 35.1%, 50.6%, and 64.1% of ICD codes needed to be adjusted for thresholds 30, 100, and 300, respectively. According to Figure 2, 100 episodes was chosen as a threshold because it represented a reasonable sample size with the clinical granularity needed, since only half of the codes needed to climb to higher level of hierarchies. For our analysis, we focused on the threshold of 100. Thus, diagnoses with less than 100 treatment episodes were considered rare situations, and ontological and rank adjustments were made for these rare events.

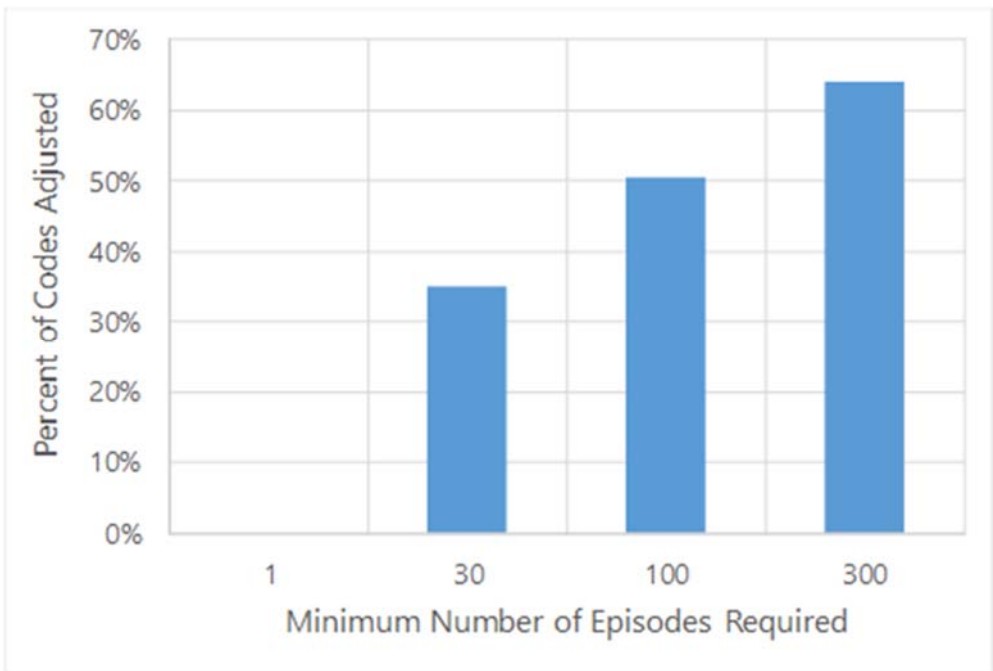

**Figure 2.** Percentage of codes adjusted for different thresholds.

### 3.2. Comparison of AROC Gaps between Models with and without OA

Table 1 shows the AROC gaps between models created with and without OA for the top 20 antidepressants in our dataset. All models lost accuracy in testing. Without the OA, the antidepressant nefazodone, a relatively less frequent antidepressant, had the highest

overfitting (Gap = 30.3%) among all antidepressants. The gaps for imipramine, fluvoxamine, nortriptyline, and vortioxetine were 29.8%, 25.7%, 21.2%, and 19.4%, respectively.

**Table 1.** AROC gaps for predicted effectiveness between the training and test sets.

| Antidepressant | Threshold 1 (No Adjustment) | | | Threshold 100 (Adjusted Data) | | |
|---|---|---|---|---|---|---|
| | **Training** | **Test** | **Gap** | **Training** | **Test** | **Gap** |
| AMITRIPTYLINE | 80.2% | 68.4% | 11.9% | 72.3% | 68.0% | 4.4% |
| BUPROPION | 73.9% | 67.3% | 6.6% | 69.8% | 67.3% | 2.5% |
| CITALOPRAM | 65.4% | 63.2% | 2.2% | 63.9% | 63.3% | 0.7% |
| DESVENLAFAXINE | 76.5% | 68.9% | 7.5% | 70.2% | 69.2% | 1.0% |
| DOXEPIN | 79.9% | 64.6% | 15.4% | 67.4% | 64.2% | 3.2% |
| DULOXETINE | 67.3% | 61.9% | 5.3% | 64.0% | 62.3% | 1.6% |
| ESCITALOPRAM | 63.7% | 60.5% | 3.2% | 61.5% | 60.3% | 1.2% |
| FLUOXETINE | 66.2% | 64.8% | 1.4% | 65.3% | 64.8% | 0.5% |
| FLUVOXAMINE | 89.2% | 63.5% | **25.7%** | 69.3% | 62.1% | **7.2%** |
| IMIPRAMINE | 94.7% | 64.9% | **29.8%** | 77.9% | 67.8% | **10.2%** |
| MIRTAZAPINE | 72.0% | 59.6% | 12.4% | 63.6% | 59.6% | 4.0% |
| NEFAZODONE | 88.7% | 58.4% | **30.3%** | 68.1% | 59.9% | **8.2%** |
| NORTRIPTYLINE | 85.1% | 63.9% | **21.2%** | 70.9% | 65.4% | **5.5%** |
| PAROXETINE | 65.0% | 62.1% | 2.9% | 62.9% | 62.1% | 0.8% |
| PRAMIPEXOLE | 76.0% | 58.6% | 17.4% | 64.0% | 58.9% | 5.0% |
| ROPINIROLE | 77.1% | 58.4% | 18.7% | 64.4% | 59.0% | 5.4% |
| SERTRALINE | 65.6% | 64.2% | 1.4% | 64.7% | 64.2% | 0.5% |
| TRAZODONE | 84.3% | 72.5% | 11.8% | 78.1% | 73.7% | 4.4% |
| VENLAFAXINE | 67.8% | 64.2% | 3.6% | 65.4% | 64.4% | 1.1% |
| VORTIOXETINE | 85.6% | 66.2% | **19.4%** | 70.3% | 68.9% | **1.3%** |

The ontological adjustment reduced the gap in AROC between training and test sets; it reduced the extent to which the parameters of the model reflected noise in the training set. For example, for bupropion, the gap in AROC was 6.6% before adjustment and 2.5% after adjustment. Nefazodone had the best performance with OA. The AROC gap of this antidepressant was reduced by 22.0%. The gaps for imipramine, fluvoxamine, nortriptyline, and vortioxetine were reduced by 19.6%, 18.5%, 15.7%, and 18.1%, respectively. On average, the gap was reduced by 9.0%, indicating that ontological adjustment reduced the extent to which noise was captured in the parameters of the model. Six models had improved test set performance (imipramine, nefazodone, nortriptyline, ropinirole, trazadone, and vortioxetine), and three performed worse (doxepin, escitalopram, fluvoxamine).

## 4. Discussion

Ontology adjustment can increase the statistical power of the analysis by aggregating rare ICD codes into broader and more frequent groupings. If conditions with less than 100 observations are considered rare, 50.6% of diagnoses were affected by our adjustments. These data suggest that nearly half of conditions in the medical history of the patient are rare, even in massive databases such as ours. When there are many rare situations in the data, the overall accuracy of analysis is affected. If we drop these rare conditions from the analysis, nearly 50% of cases are affected. With so many rare diagnoses, the overall impact of these rare events is large.

Rare events in the data cause overfitting, meaning that they increase the gap between AROC in training and test sets. This paper demonstrated that ontological adjustments can reduce the gap by 9.0%, leading to better parameter estimates from the training set. Although OA did not improve accuracy of the overall models, it reduced the gap between training and test AROCs. This means that the OA reduced the frequency of modeling noise in the training set. Therefore, the parameters estimated in these training model may be more accurate. In addition, OA increased the sample size for each parameter (especially for rare events) in the training model. Increasing sample size reduced modeling noise in

the estimation of the parameter, and modeling less noise led to AROCs that were more similar to test AROCs. Six models improved AROC on the test data, eleven had the same performance, and three performed worse. While this method did not improve quality for a majority of the models, the goal was to show a reduction in overfitting. Other algorithms to fit the data may show better performance.

The use of ontology to improve predictions is not new. In the past, investigators have combined diagnostic codes into broad categories such as Clinical Classification Software [29], Charlson Comorbidity Index [30], and Elixhauser [31]. Wei et al. [19] and Leader et al. [20] evaluated the value of grouping disease codes into broad categories. Their results showed that ICD codes were typically too detailed, while classification categories were often not granular enough. These indexes classify disease codes into a fixed small number of categories. In our paper, ICD codes were grouped into a customized category only if the number of occurrences was less than a selected threshold. Therefore, the proposed approach balances the sample size needed and clinical granularity needed. Our study confirmed that ontological adjustment increased accuracy of parameters of models, especially for parameters estimated for rare diseases. Moreover, it can be used to prevent the overfitting in big data analysis.

Our paper has several limitations. First, the dataset relied on observational data documented in administrative claims data. It was limited by the absence of symptom remission information. It lacked the reason for medication switches. While medication switches can be used to judge that the initial medication was not successful, the reverse is not always true. Second, the generalization from a single study may not be reasonable, and additional data may be needed on the effectiveness of the ontological adjustments.

## 5. Conclusions

Ontology adjustment can be utilized to group rare comorbidities into reasonably broad concepts that occur sufficiently in the data to capture the effectiveness of treatment for patients with these conditions. Ontology adjustment can increase the statistical power of the analysis by aggregating rare ICD codes into broader groupings. It also can reduce model overfitting, leading to better parameter estimates from the training set. Therefore, it can improve the accuracy for predictive models of antidepressant treatment outcomes.

**Author Contributions:** H.M. prepared the draft and provided ontology and medical coding knowledge. F.A. organized the study, obtained the funding, and participated in analysis of the data. C.A.H. and V.S.N. provided statistical direction and analysis. All authors participated in writing this paper. All authors have read and agreed to the published version of the manuscript.

**Funding:** This research was funded by The Robert Wood Johnson Foundation, grant number 76786.

**Institutional Review Board Statement:** Ethics approval was granted by the George Mason University Institutional Review Board. Patients and/or the public were not involved in the design, conduct, reporting, nor dissemination plans of this research.

**Informed Consent Statement:** Not applicable.

**Data Availability Statement:** The data used in this study are proprietary. The data are available through Optum Life Sciences in the same manner they were obtained by investigators in this study.

**Conflicts of Interest:** The authors declare no conflict of interest.

## Abbreviations

| | |
|---|---|
| ICD | International Classification of Disease |
| OLDW | OptumLabs® Data Warehouse |
| AROC | Area under the receiver operating curves |
| OA | Ontological adjustment |

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
