# Peer review of "Improving the Accuracy of Predictive Models for Outcomes of Antidepressants by Using an Ontological Adjustment Approach"

_applsci, doi:10.3390/app12031479_

Round 1

Reviewer 1 Report

The manuscript is written methodically, with adequate explanations and satisfactory use of references where appropriate.

The clarity of writing and the organization are quite good.

The technical terms are explained in detail and the topic of the paper is clear and understandable.

The presented methodology and the results are clearly communicated, with the necessary background for the readers included in the paper.

The review of the state-of-the-art is sufficient.

The novel contribution of the paper is highlighted, as well.

The conclusion section includes some discussion about the results obtained by this work and the previous works on the analysis of the same or similar data.

Author Response

Comment Reviewer 1: The manuscript is written methodically, with adequate explanations and satisfactory use of references where appropriate.  Response:   Thank you.

Comment Reviewer 1: The clarity of writing and the organization are quite good. Response:   Thank you.

Comment Reviewer 1: The technical terms are explained in detail and the topic of the paper is clear and understandable. Response:   Thank you

Comment Reviewer 1: The presented methodology and the results are clearly communicated, with the necessary background for the readers included in the paper.  Response:   Thank you.

Comment Reviewer 1: The review of the state-of-the-art is sufficient. Response:   Thank you

Comment Reviewer 1: The novel contribution of the paper is highlighted, as well.  Response:   Thank you

Comment Reviewer 1: The conclusion section includes some discussion about the results obtained by this work and the previous works on the analysis of the same or similar data.  Response:   Thank you for your comments.

Reviewer 2 Report

A Naive Bayes classifier assumes that the presence (or absence) of a particular feature of a class is unrelated to the presence (or absence) of any other feature. Authors used Naive Bayes models to predict symptom remission after taking antidepressants. Are these independence assumptions valid to this specific problem?

Have authors studied the behavior of other kind of classifiers applied to the same problem rather than Naive Bayes, such as Support Vector Machines or Neural Networks? It would be important for comparison purposes.

In Section 2.4, authors mentioned that "The best threshold for the ontological adjustment was chosen if it resulted in the best balance between the sample size (i.e., number of episodes) and levels of climbing hierarchies." However, in section 3.1 it is not clear why the threshold was set to 100, rather than the other analyzed values. Please, explain why using 100 as threshold is better than the other analyzed values, considering the established criteria of "best balance between the sample size (i.e., number of episodes) and levels of climbing hierarchies"

Please, substitute "Table 1 shows the AROC gaps between models were created with and without OA for the top..." for "Table 1 shows the AROC gaps between models created with and without OA for the top..."

Authors stated that "The ontological adjustment reduced the gap in AROC between training and test sets." Results support that statement indeed. However, comparing the AROC achieved for the test sets with and without ontological adjustment (OA), shown in Table 1, it can be seen that the OA did not significantly improve the performance. Why is that? It seems that, although the overfitting is reduced, there is no gain for generalization, which would be the most important after all. Authors should discuss that point in further details.

In section 4, authors stated "Rare events in the data cause overfitting, meaning that they increase the gap between AROC in training and validation sets." Wouldn't it be "training and test sets" instead?

In section 4, authors stated that "This paper demonstrated that ontological adjustments can reduce the gap by 8.2%, leading to better parameter estimates from the training set." However, previously, in section 3.1, they mentioned that "On average, the gap was reduced by 9.0%, indicating that ontological adjustment..." Was it 9.0% or 8.2%? Please, correct this information in the text.

Six models had improved AROC on the test data, eleven had the same performance and three did worse. While this method did not improve quality for a majority of the models, the goal was to show a reduction in overfitting. Other algorithms to fit the data may show better performance.

In section 4, authors affirmed that "Our study confirmed that ontology adjustment increases accuracy of estimating antidepressants effectiveness for patients with rare conditions." Furthermore, in section 5, they reinforce that "it can improve the accuracy for predictive models of antidepressant treatment outcome." If this is true, shouldn't the AROC of the test set of the model using OA be significantly better than the AROC of the test set of the model that does not use OA? I think the results corroborate the overfitting's reduction, but they don't allow any assumptions about accuracy improvements.

In References Section, authors should edit the format of references [20] to [27].

Author Response

Comment Reviewer 2: A Naive Bayes classifier assumes that the presence (or absence) of a particular feature of a class is unrelated to the presence (or absence) of any other feature. Authors used Naive Bayes models to predict symptom remission after taking antidepressants. Are these independence assumptions valid to this specific problem?  Response:  A naïve Bayes assumes independence.  The independence assumptions are obviously not valid in these data. Despite false assumption of independence, in large sparse data such as the data in this study, naïve Bayes has been relatively accurate in predicting patient outcomes [[1],[2],[3],[4]]. We now added “Despite false assumption of independence of predictors, in large sparse data such as the data in this study, naïve Bayes has been relatively accurate in predicting patient outcomes [25-28].” (line 132)

Comment Reviewer 2: Have authors studied the behavior of other kind of classifiers applied to the same problem rather than Naive Bayes, such as Support Vector Machines or Neural Networks? It would be important for comparison purposes. Response:  No, we have not.  This is not a study of which classifier is best but one focused on how ontological information can be used to increase sample size and reduce bias.

Comment Reviewer 2: In Section 2.4, authors mentioned that "The best threshold for the ontological adjustment was chosen if it resulted in the best balance between the sample size (i.e., number of episodes) and levels of climbing hierarchies." However, in section 3.1 it is not clear why the threshold was set to 100, rather than the other analyzed values. Please, explain why using 100 as threshold is better than the other analyzed values, considering the established criteria of "best balance between the sample size (i.e., number of episodes) and levels of climbing hierarchies".  Response:  Thank you for this comment. In the section 2.4, we now clarify our criteria as follows: "The best threshold for the ontological adjustment was chosen if it resulted in the best balance between the sample size (i.e., number of episodes) and levels of climbing hierarchies (i.e., number of affected codes)." In section 3.1, we clarify the reason for using 100 as follows: “According to Figure 2, 100 episodes was chosen as a threshold because it represents a reasonable sample size with the clinical granularity needed, since only half of the codes needed to climb to higher level of hierarchies.”

Comment Reviewer 2:  Please, substitute "Table 1 shows the AROC gaps between models were created with and without OA for the top..." for "Table 1 shows the AROC gaps between models created with and without OA for the top..."  Response:  Thank you for your comment. We removed the word “were” in line 158.

Comment Reviewer 2:  Authors stated that "The ontological adjustment reduced the gap in AROC between training and test sets." Results support that statement indeed. However, comparing the AROC achieved for the test sets with and without ontological adjustment (OA), shown in Table 1, it can be seen that the OA did not significantly improve the performance. Why is that? It seems that, although the overfitting is reduced, there is no gain for generalization, which would be the most important after all. Authors should discuss that point in further details. Response:  This is precisely why this paper is interesting.  Ontological adjustment did not improve accuracy of the overall models but reduced the difference between training and test AROCs.  To us, this means that the ontological adjustment reduced the frequency of modeling noise in the training set.  Therefore, the parameters estimated in these training model may be more accurate.  This also agrees with the fact that ontological adjustment increased the sample size for each parameter in the training model.  One conclusion is that ontological adjustment increased sample size, increasing sample size reduced modeling noise in the estimation of the parameter, and modeling less noise lead to AROC that were more similar to test AROCs.  We now explain this in detail (see second paragraph in Discussion).

 Comment Reviewer 2: In section 4, authors stated "Rare events in the data cause overfitting, meaning that they increase the gap between AROC in training and validation sets." Wouldn't it be "training and test sets" instead?  Response:  Yes. you are right.  We changed the “validation” to “test” in for consistency purposes. (line187)

Comment Reviewer 2:  In section 4, authors stated that "This paper demonstrated that ontological adjustments can reduce the gap by 8.2%, leading to better parameter estimates from the training set." However, previously, in section 3.1, they mentioned that "On average, the gap was reduced by 9.0%, indicating that ontological adjustment..." Was it 9.0% or 8.2%? Please, correct this information in the text.  Response:  Thank you for pointing this out, we have corrected it (line 188).

Comment Reviewer 2:  Six models had improved AROC on the test data, eleven had the same performance and three did worse. While this method did not improve quality for a majority of the models, the goal was to show a reduction in overfitting. Other algorithms to fit the data may show better performance.  Response:  Yes, you are right that other algorithms may improve the accuracy more.  This is not the goal of this paper and has been addressed by numerous other investigators.  Our goal is to point out that ontological adjustment makes naïve Bayes model parameters more accurate. 

Comment Reviewer 2:  In section 4, authors affirmed that "Our study confirmed that ontology adjustment increases accuracy of estimating antidepressants effectiveness for patients with rare conditions." Furthermore, in section 5, they reinforce that "it can improve the accuracy for predictive models of antidepressant treatment outcome." If this is true, shouldn't the AROC of the test set of the model using OA be significantly better than the AROC of the test set of the model that does not use OA? I think the results corroborate the overfitting's reduction, but they don't allow any assumptions about accuracy improvements. Response:  We have revised this to say that “Our study confirmed that ontological adjustment increased accuracy of parameters of models, especially for parameters estimated for rare diseases.” (line 209) The point is that without these adjustments, the parameters assigned to rare diseases come from small samples that are not reliable. 

Comment Reviewer 2:  1In References Section, authors should edit the format of references [20] to [27]. Response:  Thank you for pointing this out, we have corrected it.

[1]           Tuck MG, Alemi F, Shortle JF, Avramovic S, Hesdorffer C. A Comprehensive Index for Predicting Risk of Anemia from Patients' Diagnoses. Big Data. 2017 Mar;5(1):42-52.

[2]           Min H, Avramovic S, Wojtusiak J, Khosla R, Fletcher RD, Alemi F, Elfadel Kheirbek R. A Comprehensive Multimorbidity Index for Predicting Mortality in Intensive Care Unit Patients. J Palliat Med. 2017 Jan;20(1):35-41.

[3]           Alemi F, Levy CR, Kheirbek RE. The Multimorbidity Index: A Tool for Assessing the Prognosis of Patients from Their History of Illness. EGEMS (Wash DC). 2016 Oct 13;4(1):1235.

[4]           Tabatabaie M, Sarrami AH, Didehdar M, Tasorian B, Shafaat O, Sotoudeh H. Accuracy of Machine Learning Models to Predict Mortality in COVID-19 Infection Using the Clinical and Laboratory Data at the Time of Admission. Cureus. 2021 Oct 14;13(10):e18768. doi: 10.7759/cureus.18768. PMID: 34804648; PMCID: PMC8592290.

Round 2

Reviewer 2 Report

The authors have satisfactorily addressed most of my concerns. I asked authors to include studies with other classifiers applied to the same problem, rather than Naive Bayes, such as Support Vector Machines or Neural Networks. However, they did not, because they said "This is not a study of which classifier is best but one focused on how ontological information can be used to increase sample size and reduce bias." Although, I still think that it would be interesting to see how ontological information contributes for the performance of different classifiers, I consider it is not mandatory for the paper acceptance. So, I recommend the acceptance of the paper in its present form.